# A Spatio-Temporal Ensemble Deep Learning Architecture for Real-Time Defect Detection during Laser Welding on Low Power Embedded Computing Boards

**DOI:** 10.3390/s21124205

**Published:** 2021-06-18

**Authors:** Christian Knaak, Jakob von Eßen, Moritz Kröger, Frederic Schulze, Peter Abels, Arnold Gillner

**Affiliations:** 1Fraunhofer-Institute for Laser Technology ILT, Steinbachstrasse 15, 52074 Aachen, Germany; jakob.von.essen@ilt.fraunhofer.de (J.v.E.); frederic.schulze@ilt.fraunhofer.de (F.S.); peter.abels@ilt.fraunhofer.de (P.A.); arnold.gillner@ilt.fraunhofer.de (A.G.); 2Faculty of Mechanical Engineering, Chair for Laser Technology, RWTH Aachen University, Steinbachstrasse 15, 52074 Aachen, Germany; moritz.kroeger@llt.rwth-aachen.de

**Keywords:** real-time process monitoring, recurrent neural network, high-speed infrared imaging, convolutional neural network, lack of fusion (false friends), feature importance, AI edge device

## Abstract

In modern production environments, advanced and intelligent process monitoring strategies are required to enable an unambiguous diagnosis of the process situation and thus of the final component quality. In addition, the ability to recognize the current state of product quality in real-time is an important prerequisite for autonomous and self-improving manufacturing systems. To address these needs, this study investigates a novel ensemble deep learning architecture based on convolutional neural networks (CNN), gated recurrent units (GRU) combined with high-performance classification algorithms such as k-nearest neighbors (kNN) and support vector machines (SVM). The architecture uses spatio-temporal features extracted from infrared image sequences to locate critical welding defects including lack of fusion (false friends), sagging, lack of penetration, and geometric deviations of the weld seam. In order to evaluate the proposed architecture, this study investigates a comprehensive scheme based on classical machine learning methods using manual feature extraction and state-of-the-art deep learning algorithms. Optimal hyperparameters for each algorithm are determined by an extensive grid search. Additional work is conducted to investigate the significance of various geometrical, statistical and spatio-temporal features extracted from the keyhole and weld pool regions. The proposed method is finally validated on previously unknown welding trials, achieving the highest detection rates and the most robust weld defect recognition among all classification methods investigated in this work. Ultimately, the ensemble deep neural network is implemented and optimized to operate on low-power embedded computing devices with low latency (1.1 ms), demonstrating sufficient performance for real-time applications.

## 1. Introduction

Process monitoring and fault detection are an essential requirement for a multitude of manufacturing processes and are particularly relevant and necessary when human safety is at stake (e.g., safety-critical automotive parts, battery parts, aerospace parts). In addition, however, for joining processes in the process industry or in power generation (e.g., nuclear power plants), early detection of defects and deviations can shorten processing time and enable online compensation of process deviations for “first time right” production. In particular, complex joining processes such as laser welding (LW) require suitable quality monitoring procedures in order to satisfy the constantly increasing demands for high-quality products in modern and flexible production environments. In laser deep-penetration welding, a laser beam is focused on the material’s surface. The energy provided by the laser radiation heats the welding material and as a result, the temperature at the focus of the laser beam exceeds the boiling point of the material. This leads to a vapor capillary (keyhole) which increases the penetration depth of the laser beam into the material due to the occurrence of multiple reflections within the keyhole. Although laser welding processes are well known, automated in-line quality diagnosis still remains a challenge [1]. In practice, weld quality is affected by several factors, such as thermal conditions during laser–material interaction, variations in material properties, impurities on the workpiece surface, and changes in the properties of the laser beam, all of which may result in an unacceptable product [2,3]. During laser welding the complex interaction between laser beam and the weld material can lead to weld imperfections such as cavities, solid inclusion, lack of fusion as well as lack of penetration, weld seam deformations, cracks, and other deviations from the desired weld quality. A reliable quality diagnosis tool must provide high sensitivity for critical defects but also a certain adaptability in case of required process changes.

A common method for monitoring a laser welding process is to observe the radiation emitted by the keyhole via high-speed photodiodes. The keyhole is an out-gassing channel for vaporized material and process gases. As a result of the outflowing gases and the incoming laser radiation, a weld plume originates above the material’s surface on the keyhole position. With respect to in-process monitoring, the electromagnetic signature of the keyhole and weld plume can be observed and correlated with quality-related phenomena, which occur during the weld process [4,5]. Unfortunately, the correlations of those signals to certain quality criteria are often ambiguous, so that statistical proof of quality by destructive testing is necessary.

However, recent advances in sensing technology and an increasing number of sensors applied on laser machines and processes enable online weld quality monitoring with higher precision by combining multiple data sources. Similarly, complex sensors such as thermal camera systems have become reasonably priced and can be used as a data source for in-process weld quality monitoring. Recently used sensors for laser welding process monitoring are image-based sensors such as cameras in the infrared wavelength range [6], acoustic emission sensors, optical sensor such as high-speed photodiodes and pyrometer [7]. Furthermore, techniques such as x-ray imaging, spectrographically sensors [8] and combined sensing techniques have been investigated [9]. Especially, camera sensors provide important information from various process zones that emerge during laser welding. The keyhole is typically surrounded by molten material, the weld pool. Size and shape of weld pool are important geometrical parameters that correlate with weld shape and quality [10,11].

Due to high process dynamics and partially chaotic keyhole behaviors [12], an approach based on precise physical modelling of the welding process is not practical for real-time quality diagnosis of laser welds [13]. On the other hand, the incorporation of new technologies such as Industrial Internet of Things (IIoT) and advanced analytics into manufacturing systems aims to produce individualized products at high quality and low costs. In the manufacturing domain, such data-driven approaches have been extensively studied in the past and are based on autoregressive (AR) models, cluster analysis, fuzzy set theory or on supervised learning algorithms such as multivariate regression, multi-layer perceptron and decision trees, as well as k-nearest neighbors [14,15]. Therefore, recent development led to advanced process monitoring systems which integrate machine learning techniques for process control and prediction of critical defects [16,17]. An advantage of data-driven methods is that it is not necessary to explicitly model the physical behavior of the system in order to build a statistical model. However, process understanding can help to design and develop the right feature set and to select relevant sensors and signal sources as input for the data-driven model. A data-driven model utilizes input variables (features) extracted from the raw signals to establish a statistical model between those features and the observed phenomena, e.g., weld defects during the welding process. Therefore, features that describe the significant characteristics of the signal are required for classical supervised learning algorithms and are often manually designed and depend on signal type (e.g., image data, data from high-speed photo diodes) and the output variable. For example, You et al. [18] proposed a diagnosis system for autonomous laser beam welding. This system is based on extracting features with wavelet packet decomposition and dimensionality reduction techniques (PCA) in combination with SVM-based classification for defect detection. An extensive experimental setup has been established to evaluate the proposed methods comparing measurements signals from photodiodes, image sensors and x-ray analysis. However, the question remains which features are necessary to achieve high defect detection accuracies and how different learning algorithms may improve the detection performance.

From the field of computer vision and pattern recognition, deep learning methods have emerged as an effective technique to solve signal- and image-processing tasks [19,20]. Deep learning is different from classical machine learning as it integrates the process of feature extraction within the data-driven model. Deep learning models with multiple layers of artificial neurons are based on the findings in neuroscience that multi-stage deep neural networks allow humans to perform complex signal processing tasks such as object and voice recognition [21,22]. As a result, deep learning models are capable of extracting more refined and complex image characteristics and are therefore expected to provide higher classification accuracies than conventional approaches based on feature engineering and traditional classifiers. With the advent of deep learning, especially convolutional neural networks (CNN), top rankings in classification performance were achieved in several image recognition competitions such as ImageNet in 2012. CNNs have therefore become a common solution for many computer vision tasks [23]. Nowadays, it is possible to train large multilayered CNN networks, typically consisting of many types and numbers of layers on GPU-hardware, with the help of open source deep learning frameworks such as TensorFlow [24], PyTorch [25] or Caffe [26].

This has led to various applications of CNNs in industrial production sector to recognize defects and improve product quality [27,28,29,30]. Therefore, it is no surprise that deep learning has recently been used in laser welding applications to predict defects [11,31].

For example, in 2014, Günther et al. [32] suggested a deep learning scheme for extracting relevant features from in-process laser welding data. They used a deep learning-based auto-encoder with fully connected layers to create a new latent feature space of 16 features that describe the welding images.

Thermal images and convolutional neural networks work well in combination, as shown by Gonzales-Val et al. [33]. The authors proposed a CNN architecture to predict dilution in laser metal deposition as well as defects in laser welding based on infrared images. First experiments show promising results with respect to the prediction accuracy. For CO2 laser welding, a combination of CNN and a recurrent neural network (RNN) was applied to extract primary features from weld pool images. Although RNNs are used to model sequence-based problems such as voice recognition, in this approach, RNNs were used to fuse features extracted via CNN from a single image with the help of a RNN to recognize good and imperfect weld images [34].

In addition to the manufacturing domain, architectures based on CNN and RNN turned out to be successful in applications such as action and emotion recognition in video data [35,36] and wearable activity recognition [37]. Additionally, a group of researchers utilized CNN and RNN architectures to improve prediction accuracy of the steering angle of an autonomous vehicle. They achieved the lowest error compared to other approaches in the literature [38].

In this study, a new ensemble deep learning approach for data-driven feature extraction and weld defect detection is introduced and investigated. The architecture is based on convolutional neural networks (CNN) which are often used for image classification and is further described in Section 2.2. Although in-process data are available in the form of images, some important information may only be available in the time domain of the welding video stream. Therefore, the CNN is combined with a recurrent neural network (RNN), specifically the gated recurrent unit (GRU) architecture as described in Section 2.3, that was recently used to solve pattern recognition tasks in the time domain [39]. The advantage of CNNs to extract relevant spatial information and the ability of GRUs to learn meaningful temporal characteristics are combined to automatically extract a spatio-temporal feature representation of a given image sequence. Furthermore, high performance classification algorithms, namely, k-nearest neighbors (kNN) and support vector machines (SVM) are used to build an ensemble deep learning model with the goal to obtain better generalization performance and higher classification robustness.

The ensemble deep learning framework is compared with established architectures such as ResNet50V2 [40], MobileNetV2 [41] and InceptionV3 [42] and a baseline CNN architecture without GRU layer and the ensemble strategy. In addition, a comparison of the proposed framework with conventional machine learning using manual feature extraction methods is given. For that, geometrical and statistical features are extracted from thermal image data (MWIR and NIR) to determine the keyhole and weld pool characteristics for each time step. The features are based on higher-order image moments, shape descriptors and descriptive image statistics as well as statistics in the time domain, that are used to establish a high-dimensional feature vector. In addition, we determine the significance of individual features and the relevance of different feature subsets in terms of their classification performance. Subsequently, all models are optimized using a grid search process combined with nested cross validation. In a further step, the deep learning architectures are compared with classical machine learning approaches based on the individual prediction performance in four unseen welding trials. Finally, the proposed ensemble deep learning model is optimized for real-time inference as well deployed and evaluated on an embedded computing board (NVIDIA Jetson AGX Xavier). A schematic overview of the data processing and evaluation steps applied in this work is given in Figure 1.

Overall, the main contributions of this work include the following points:Development and evaluation of a unique ensemble deep learning architecture combining CNNs and GRUs with high performance classification algorithms (i.e., SVM, kNN) for real time detection of six different welding quality classes;Comparison of the proposed architecture with available deep learning architectures as well as classical machine learning methods based on manual feature extraction;Assessment of the significance of geometric and statistical features extracted from the keyhole and weld pool region of two different image data sources (i.e., MWIR and NIR) with respect to the ability to detect particular weld defects;Development and evaluation of a real-time inference pipeline for the proposed method operating on low-power embedded computing devices.

From here, the remaining part of this paper shows the following structure. Section 2 provides the background knowledge for different classification algorithms as well as a definition of the proposed ensemble CNN-GRU architecture. Section 3 describes the experimental setup and the process of feature extraction. Experimental results are presented and analysed in Section 4. Finally, a conclusion is given in Section 5.

## 2. Methodology and Background Knowledge

In this work, several conventional machine learning algorithms are compared in terms of prediction performance and processing time. These algorithms and the resulting prediction model often require feature engineering as a preliminary stage, especially in the field of image recognition, in order to create predictive models not only with a high prediction performance and less overfitting, but also with a high degree of comprehensibility. The investigated conventional machine learning algorithms are listed below:Decision tree (DT);K-nearest neighbors (kNN);Random forest (RF);Support vector machine (SVM);Logistic regression (LogReg);Artificial neural network (ANN).

A detailed overview and discussion of these algorithms can be found in several textbooks such as [43,44,45,46]. The method of feature engineering and classification using conventional algorithms is additionally compared to modern deep learning approaches, which include the process of feature extraction as part of the model. Both conventional and deep learning approaches use the following data set D as input to establish a data-driven model:(1)D={(xi,yi)∣xi∈ℝp,yi∈ℝm,i=1,2…,Q}
where xi denotes the ith feature vector, which for conventional machine learning methods consists of numerous features p, that are explained in Table A1 and Table A2 in further detail. For deep learning algorithms, the feature vector xi represents a raw image or image sequence in the data set. The label vector, described by yi, belongs to the feature vector xi while m denotes the number of classes, which in this work represents the six different welding quality states as stated in Section 3.3. For this study, the DT, LogReg, SVM, ANN, kNN and RF implementations of scikit-learn 0.22.1 and Python 3.6 were used to train classification models [47]. All hyperparameters that were optimized via grid search and 4-fold nested cross validation can be obtained from Table A3. For all other hyperparameters not listed in Table A3, the default values of the scikit-learn implementation are used. In the subsequent section, a more detailed description regarding the combination of CNN and GRU architectures used in this work is given.

### 2.1. Convolutional Neural Network (CNN)

CNNs can not only be used for image data, but they bring certain advantages to these applications, such as translation invariance through weight sharing, and local connectivity that takes the spatial structure of images into account. For some other applications, where spatial relations are important, these CNN model assumptions of may also be applicable.

A simple CNN usually consists of three types of layers, which are stacked to create a deep neural network model. These layers are usually defined as pooling layer, fully connected layer and convolutional layer. In the convolutional layer, small patches (filter) convolve over the input array, which in the first convolutional layer is the original image. The coefficients of each filter kernel defined in a certain convolutional layer are determined during the training process. The output of a convolutional layer can be denoted as follows [48]:(2)$$X_{d}^{l}=f\left(\sum_{i\in M_{d}}X_{i}^{l-1}\times K_{id}^{l}+b_{d}^{l}\right).$$
where $X_{d}^{l}$ is the dth output feature map (image) of the lth convolutional layer. On the right side, the ith output feature map $X_{i}^{l-1}$ of the previous layer l−1 is convolved with the idth kernel K of the current layer. $b_{d}^{l}$ denotes the offset (bias), and $M_{d}$ represent the input feature maps while f represents the activation function.

The convolutional layer is frequently followed by a pooling layer to reduce the input dimensions for the following layers by down-sampling feature maps from the previous layer. Typical types of pooling layers are max pooling and average pooling. The output $x_{d}^{l}$ is stated by the following equation:(3)$$X_{d}^{l}=f(\delta_{d}^{l}\;\mathrm{subsample}\left(X_{d}^{l-1}\right)+b_{d}^{l}$$
where l is the number of the pooling layer, f can be an activation function, $\delta_{d}^{l}$ denotes the resample factor and subsample(.) represents the down-sampling function (e.g., mean or max pooling), and $b_{d}^{l}$ is the bias (offset). Pooling, especially max pooling, is a convolution-based operation that is applied to reduce overlapping in feature maps and can help to avoid overfitting and may lead to a more generalized model [19].

### 2.2. Recurrent Neural Networks and Gated Recurrent Units (GRU)

In this work, CNNs are utilized to automatically extract relevant characteristics from raw camera images. It is also possible to extract spatio-temporal information from video streams using 3D-CNNs, to extract patterns from temporal changes between adjacent frames. For example, 3D-CNNs are often used to recognize gestures or emotion in videos [35,36,49]. However, compared with approaches that combine CNN with RNN structures such as long short-term memory (LSTM) or gated recurrent units (GRU), 3D-CNN has a disadvantage that derives from its high computational complexity and excessive memory consumption, which can be a major burden for several applications that require high inference rates, especially on embedded devices [50]. Additionally, RNN architectures can be used to extract long-term temporal characteristics, whereas 3D-CNNs are mostly used for the extraction of short-term temporal pattern [51]. Therefore, the combination of CNN and LSTM has been used recently for action recognition in video data that is still a challenging problem in computer vision [34,52]. LSTMs have become especially popular due to high performances achieved in domains such as natural language processing, but recent findings suggest that GRU architectures offer very comparable accuracies compared to LSTM with lower computational costs. [39,53].

GRUs were proposed by Cho et al. [54] in 2014 as an alternative architecture to the commonly used long short term memory (LSTM), which was proposed in 1997 [55]. The GRU is a slightly more simplified variation of the LSTM, as it has fewer parameters and thus may train faster and needs less data to generalize. Compared to LSTM, the entire memory is exposed to the network, while for LSTMs the exposure to other units is controlled by the output gate. Additionally, GRU can control the information flow from the previous activation, whereas LSTM is not able to manage this information flow [53]. Potentially lower calculation costs and the data-efficient structure are the reason why GRU is used for this work. The main advantage is that gated units in RNNs can store information in their units that is accessible in a later time step. The decision when to store, read or erase information is learned from the data. A GRU with unit u in layer l can be described as follows [56]:(4)$${\tilde{h}}_{l,u}^{t}=g_{1}\left(w_{l,u}\;x^{t}+u_{l,u}h_{l}^{t-1}+b_{d,u}^{l}\right)$$
(5)$${ℶ}_{l,u}^{t}=g_{2}\left(m_{l,u}\;x^{t}+o_{l,u}h_{l}^{t-1}+c_{d,u}^{l}\right)$$
(6)$$h_{l,u}^{t}={ℶ}_{l,u}^{t}{\tilde{h}}_{l,u}^{t}+\left(1-{ℶ}_{l,u}^{t}\right)h_{l,u}^{t-1}$$
(7)$$h_{l}^{t}=\left[h_{l,1}^{t-1},\ldots ,h_{l,n\_unit}^{t-1}\right]$$
(8)$$y_{l}^{t}=g_{3}\left(V_{l,}h_{l}^{t}+a_{d,u}^{l}\right)$$
where the parameter vectors $w_{l,u}\;x^{t}$, $u_{l,u}$,, $m_{l,u}$,$\;o_{l,u}$ and $V_{l,}$ as well as the parameter $b_{d,u}^{l}$, $c_{d,u}^{l}$, $a_{d,u}^{l}$ are determined during the training via backpropagation through time. $g_{1}$ represents the tanh activation function and $g_{2}$ is implemented as sigmoid function. If the gate value ${ℶ}_{l,u}^{t}$ is close to zero, the GRU keeps the state values $h_{l}^{t-1}$, but saves a new state ${\tilde{h}}_{l,u}^{t}$ if the gate value is close to 1. The input of the GRU is a feature vector $\;x^{t}$ at time step t and a vector $h_{l}^{t-1}$ that contains state values from all units in the previous layer. $g_{3}$ is an activation function and is represented in this work by the sigmoid function. In our architecture, the feature vector extracted by the CNN is consecutively fed into the RNN layer, which is represented by a GRU. The overall CNN-GRU architecture is shown in Figure 2.

For each measurement, the network takes a sequence of $n_{sequence}$ consecutive weld images as input. Instead of using only the most recent image, the network is able to use information from the last $n_{sequence}$ images to predict the local weld quality. The image sequence represents the input of the first convolution layer, where convolution kernels with a size of 2×2 are applied on the input images. Based on Equation (2), this results in a specific number of feature maps defined by the hyperparameter conv_1_depth. A second convolution layer uses the previously calculated feature maps as input and convolves a 3×3 kernel to compute the second layer feature maps with the help of the activation function (Activation), number of feature maps conv_2_deph and Equation (2). The results are transmitted to the pooling layer that applies maximum pooling on each feature map, where a kernel of size 2×2 moves with a step size of 2 in both directions (Equation (3)).

The GRU network is implemented at the end of the convolutional stack of the network. The flattened feature maps (i.e., 25 × 2352 matrix) of the 25 images are used as input for the GRU layer that consists of a specific number of units (GRU_units) that use tanh activation function. Based on Equation (8), the GRU layer combines the feature vectors of a sequence of $n_{sequence}$ consecutive weld images to obtain a spatio-temporal feature representation.

The last fully connected layer represents a hidden layer that consists of a specific number (Dense_units) of nodes and uses the activation function (Activation). The softmax function is selected as the activation function of the last output layer.

Additionally, a reference CNN was trained based on a modified architecture compared to Figure 2, that uses a single image as input and has no GRU layer (i.e., CNN-baseline). For both architectures, hyperparameters such as depth of each convolutional layer (conv_1_depth) and (conv_2_depth) as well as the activation function (Activation), the number of units of GRU layer (GRU_units), the number of units in the fully connected layer (Dense_units) and the length of the input image sequence $n_{sequence}$ were determined via grid search on the basis of the values provided in Table A3. For each training process, Nesterov-accelerated Adaptive Moment Estimation (Nadam) optimizer was used to minimize the categorical cross-entropy loss function within 100 training epochs.

### 2.3. Ensemble Deep Learning

In ensemble learning, several base models are trained, and the individual outputs are aggregated using a decision fusion strategy to increase the generalization capabilities and the robustness of the final model [57].

In this study, we first train the CNN-GRU architecture with two dense layers as classification head. In a further step, the spatio-temporal image features extracted from the CNN-GRU layers are used as input to build a SVM and a k-nearest neighbors classification model. Finally, the unweighted average of the individual predicted class probabilities is used to determine the final class.

The ensemble deep learning architecture is also compared to state-of-the-art CNNs for image processing. For this purpose, common architectures such as ResNet50V2 and MobilNetV2 and InceptionV3 are used for comparison with respect to prediction performance and inference times. By replacing the original classification heads, the pre-trained models are trained with new fully connected classification heads consisting of a hidden layer (i.e., 768 nodes) and the output layer (i.e., 6 nodes). In this study, all other layers except the new dense layers were set to be untrainable.

All deep learning architectures in this work were implemented using TensorFlow 2.3 and Python 3.6.

## 3. Experiment Setup and Data Preprocessing

### 3.1. Multi-Camera Welding Setup

In order to detect changes in process conditions and quantify process imperfections, online process monitoring based on two cameras, as shown in Table 1, was applied. A CMOS-based camera (NIR) was used to visualize the keyhole and its surrounding area during the welding process. To monitor the weld pool in real time, a PbSe-sensor (MWIR) was engaged, since the maximum of temperature radiation occurs according to Equation (9) within the wavelength range of the sensor’s sensitivity. The relation between a specific temperature and its wavelength of maximum thermal radiance can be expressed by the following equation according to Wien’s displacement law [58]:(9)$$\lambda_{max}=\frac{2897.8\;\mu m\;\times K}{T_{melt}}$$

Substituting $T_{melt}$ by a value of 1737 K, which represents the melting point of low carbon steel (FE P05) used for these experiments, leads to the wavelength of maximum thermal radiance at $\lambda_{max}=1634\;\mathrm{nm}$. In front of the camera sensor, narrow bandpass filters reduce the effect of chromatic aberration on the measurement signal. To meet the $\lambda_{max}$ calculated above, the infrared camera uses a filter that provides a bandwidth of 82 nm at a central wavelength of 1690 nm, as shown in Table 1. Both cameras start capturing image data when triggered by a signal from the robot control system. However, the data acquisition rates of the cameras used for this experiment differ. Considering the MWIR-camera sample rate of 500 Hz, each frame of the NIR-camera (100 Hz) is multiplied by 5 to avoid down-sampling of the 500 Hz signal and to synchronize the data streams.

Experiments have been conducted by applying different welding parameters using a high-power disk laser at a focus diameter of 0.6 mm and argon as shielding gas. The experiment was performed with galvanized low-carbon steel in overlapping configuration. The geometric dimensions can be obtained from Figure 3.

A welding configuration, which consisted of three galvanized steel sheets (FE P05) of different thickness, was considered for the experiment. For some welding trials, a modified middle sheet was used to provoke lack of fusion in certain areas due to a larger gap size, as shown in Figure 3b. To allow outgassing of vaporized zinc during welding, a gap of 0.15 mm was established between all welding sheets.

### 3.2. Feature Extraction for In Situ Weld Image Data

This chapter describes the features being extracted from the MWIR and NIR image data that were recorded during welding processes. As stated above, the recorded video data of the welding processes contain spatio-temporal information regarding the optical emission of the weld pool and the keyhole. While the proposed deep learning approach extracts relevant features directly from the raw input data, conventional classification algorithms investigated in this work require the extraction of handcrafted features from the original data as input to work properly. Overall, 172 unique features are extracted from the two process image types shown in Figure 4 to reduce the amount of data to be processed and to counteract the effect of overfitting when using the raw images as input. The process of feature extraction is based on the following image processing steps:Binarize image based on the target object threshold (keyhole threshold > weld pool threshold);Detect contour (connected boundary line of an object) using the algorithm of Suzuki et al. [59] and select largest contour from all contours found in image;Calculate contour properties such as centroids and other image moments Table A1);Fit an ellipse to the found contour;Obtain geometrical parameters of the ellipse (Table A1 in Appendix A);Calculate additional features such as statistical and sequence-based features (Table A2).

Taking into account two different image types (i.e., NIR and MWIR image data), 86 features are calculated for every ith image and for each image type T. Equation (11) shows the aggregated feature list $F_{i}^{T}$ which consists of several feature subgroups as stated in Table 2. Geometrical features $G_{i}^{T}$ based on the extracted keyhole and weld pool contours are defined as one feature subgroup.
(10)$$F_{i}^{T}=G_{i}^{T}+IS_{i}^{T}+TS_{i}^{T}+WP_{i}^{T}+KH_{i}^{T}$$

Additionally, features related to overall images statistics such as *mean*, *minimum*, *maximum*, *variance*, *median*, *skewness* and *kurtosis* define the second subgroup $IS_{i}^{T}$. Furthermore, features based on the statistics of pixels within the keyhole region $KH_{i}^{T}$ or the weld pool area $WP_{i}^{T}$ are also defined as feature subgroups. Additionally, features are extracted from the time domain of the welding video data to form the feature subset $TS_{i}^{T}$. To this end, statistics are calculated according to Table A2, based on the weld pool area of the nine most recent consecutive images, including the current image for each time step. If no image is available for a particular position in the sequence, the values are subsequently filled with the previous value.

To improve the classification performance and robustness of the trained models, feature normalization was applied for both handcrafted features and raw image data. The following equation normalizes the features to a value between zero and one:(11)$$x_{norm}=\frac{x-x_{min}}{x_{max}-x_{min}}$$

### 3.3. Welding Defects and Data Preparation

In a further step, several welding trials based on zinc-coated steel sheets were manually characterized by human experts in terms of quality according to international standards (i.e., EN ISO 13919-1/EN ISO 6520-1) [60,61].

Figure 5 shows examples of MWIR images of different weld quality states investigated in this work. It is also shown that the amount of labeled data available for supervised learning differs greatly between defect classes. Naturally, labels for images showing a satisfactory weld situation are abundant, while image data related to small defects within the weld are rare. There are examples of different weld defects such as lack of fusion, which often appears as a good weld, in the top view, while the cross-sectional view shows a missing connection between the two sheets as shown in Figure 6. It can be obtained that sagging or an irregular weld width can easily be recognized from the top view photography. However, additional information is required to distinguish the classes of sound weld from lack of fusion and lack of penetration. To generate annotations for supervised machine learning, the image data were compared with the weld seam photography (top/bottom view) and the associated metallographic characterization (cross-sectional view) by matching both data sets via process start and end points. Only image data for which the quality of the weld seam could be reliably determined were annotated accordingly.

Overall, 14,530 images were manually annotated based on 13 weld trials. To form a temporal data set for the CNN-GRU architecture, 25 consecutive images and the associated quality labels are taken in the original temporal order. The last quality label of the image stack is used as a label for a new temporal sample to build a new data set. After moving on from one image in the original data set, the next 25 images and the corresponding label are taken and then added to the new data set. In case not all 25 images are available, the missing images are filled with the last available image. Finally, the new data set contains as many samples as the original one, but each sample consists of 25 images instead of one.

In this work, deep learning models utilize data augmentation to artificially increase the data set to 72,650 images and image sequences. Some weld trials were performed in different directions compared to the sensor alignment (e.g., Figure 5—Irregular width). To learn features that are independent of the welding direction (or the sensor alignment), image augmentation is performed for all images and image sequences. Mirroring and rotation were chosen because they allow the convolutional structures to learn rotational and directionally invariant features, which leads to a more generalized model [62]. Deep learning methods typically require more data since they come with an increased number of parameters to be trained compared to conventional methods [63]. For this welding data set, experiments have shown that with an increased amount of training data, an increase in performance can be achieved. Data augmentation was also used for image sequences. In this case, all images in the sequence were coherently augmented by using the same method (i.e., rotation, mirroring) for each image in the stack. Data augmentation was not applied to the classical methods, because most of the extracted features do not vary with image mirroring or rotation.

## 4. Results and Discussion

The next section presents the feature evaluation, the results of the comparison among the classification algorithms and the final performance evaluation based on complete and unseen weld trials. Various metrics can be used for assessing the performance of classification models. Accuracy, for example, has the advantage of being simple to interpret as it represents the ratio of correctly classified samples to the number of total samples. However, accuracy is not considered a robust measure when dealing with unbalanced classes, which is the case for the weld data set. Therefore, the F1-Score is introduced as main metric to measure multi-class classification performance on the unevenly distributed weld data set [46]. On the basis of the definition of true positive (TP), true negative (TN), false positive (FP) and false negative (FN) detections, accuracy and F1-Score are defined as follows [43]:(12)$$\mathrm{Accuracy}=\frac{\mathrm{TP}+\mathrm{TN}}{\mathrm{TP}+\mathrm{TN}+\mathrm{FN}+\mathrm{FP}}$$
(13)$$\mathrm{Recall}=\frac{\mathrm{TP}}{\mathrm{TP}+\mathrm{FN}}$$
(14)$$\mathrm{Precision}=\frac{\mathrm{TP}}{\mathrm{TP}+\mathrm{FP}}$$
(15)$$F_{1}=2\times \frac{\mathrm{Precision}\times \mathrm{Recall}}{\mathrm{Precision}+\mathrm{Recall}}$$

Accuracy score usually is utilized when the true positives and true negatives matter more, whereas F1 score typically applies when the false negative and false positive predictions are more important. In this study, accuracy and F1-Score are reported to describe the classification performance of a machine learning model, however, the F1-Score is considered for final evaluation.

### 4.1. Assessment of Feature Importance

The importance of the features was determined using sequential forward floating selection (SFFS), which represents an extension of the simpler SFS algorithm. SFS starts with an empty feature subset and trains a classification model for each available feature based on a defined algorithm, which in this case was a linear SVM classifier, as it provides short training and inference times.

The feature that provides the highest balanced accuracy score is included as the most important feature in the new subset. Afterwards, at every ith step, classifiers are trained for each combination of the (i−1)th important feature and the remaining features to determine the ith most important feature. The floating version of SFS (SFFS) has an extra step that allows the removal of features that were previously included (or excluded), resulting in an increased search space to find the optimal feature subset. It has been shown that SFFS enables the selection of appropriate features with high efficiency, especially compared to methods such as “*Min-Max search*”, “*branch and bound*” or SBS, which is why it is used in this work [64]. Based on the SFFS algorithm, Figure 7 shows the cumulated accuracies for 20 out of 172 features based on weld pool and keyhole characteristics, as well as overall image and time series statistics extracted from MWIR and NIR welding images. Starting with the far-left feature, SVM models were trained and evaluated consecutively by adding a feature in a further step, until 20 features were added to the final feature subset.

Figure 7 shows that among the five most important features, only one feature is related to the NIR camera. Interestingly, even though both cameras are imaging the keyhole region as shown in Figure 4, features of the MWIR camera are ranked as more important, probably due to the higher dynamic range of the camera. The figure further shows that at least 15 features are required to train classification models with accuracies equal to or greater than 97.8%. It can also be observed that statistics of the weld pool pixel distribution and that of the keyhole axis (i.e., *MWIR_keyhole_axis_x*/*y_kurtosis*) and further geometrical properties such as keyhole contour moments (i.e., *3^rd^_order_mom[M03|µ00]*) are most relevant for weld defect prediction. Additionally, time series features from the keyhole and weld pool region (i.e., *MWIR_keyhole*/*ts-area_variance*) also appear among the top ten features. Table 3 shows the defect detection performance of several feature subsets derived from the original amount of 172 features as cross-validated F1-Score.

The results show that feature subsets based on geometric features (MWIR+NIR (*geometrical*)) extracted from the weld pool and keyhole regions can almost reach the top F1-Score of 0.978 achieved by the original feature set. Interestingly, if the prediction model is trained only on geometrical features from either MWIR or NIR images, its performance (0.928 and 0.826) is significantly lower than the performance of the combination of these feature subsets (0.970). The general performance level of feature subsets only based on overall image statistics (*image stats*) and time series statistics (*timeseries stats*) is low compared to all other subsets. One reason for that can be found in the low dimensionality of those subsets (i.e., six features). Meanwhile, the F1-Scores for weld pool features extracted from MWIR and MWIR plus NIR images are 0.966 and 0.969, respectively, whereas the score for weld pool features extracted from the NIR images is 0.918. This is probably caused by the low thermal signal obtained with this sensor. Although NIR image data at 840 nm wavelength provide higher spatial resolution of the keyhole area, the thermal signal of the weld pool area was hardly detected by this sensor. As explained in Section 3.1, the optimal wavelength for weld pool observation is located at around 1634 nm, which is preferably observed by the MWIR camera. Additionally, if the performances of features extracted only from MWIR images are compared, weld pool features outperform the keyhole feature by 3.3%.

Overall, most relevant information can be found in MWIR features which reach, according to Table 3, generally higher F1-Scores compared to NIR features. However, the highest F1-Scores are achieved by combining features from both cameras. This leads to the assumption that the NIR images with spatially higher resolution can provide additional information of the keyhole area, compred to that obtained from the MWIR images. However, as the number of features used to create a classification model increases, the risk of overfitting also increases.

Although deep learning and, subsequently, CNN-architectures are often considered a black box model, visualization of layer-wise activation maps can provide useful information for understanding how successive intermediate convolutional layers transform their input. It also offers a first idea of the meaning of the learned filter properties and which image regions might be especially important for distinguishing the weld defects.

In Figure 8, activation maps for the first convolutional layer are shown. For different input images the activation map for each filter learned during the training is shown. The left column shows the activation maps evoked by a process of a sound weld image and the second column shows the activation maps based on several images showing different process deviations. The differences of the filter maps are depicted in the right column. The feature maps show different results depending on the camera image that was given as input. For example, irregular width images lead to different keyhole and weld pool shapes, as shown in the first row of Figure 8, whereas lack of fusion images provoke great differences in activations in the keyhole area.

Overall, each defect shows its own fingerprint in the form of activation maps extracted from the CNN-model. The similarities between the activation maps for the defects of lack of fusion and sagging result in the model’s tendency to confuse these classes. This result is also supported by the generally lower F1-Scores for this type of defect, as shown in Table 3. In case of lack of fusion, sagging or lack of penetration, most activation maps show high activity in the keyhole area and its immediate surroundings. Additionally, when the defects irregular width and sagging occur, areas related to the weld pool are activated.

### 4.2. Model Comparison Based on Grid Search Results

For a comprehensive comparison of different classification methods and algorithms, a grid search coupled with 4-fold nested cross validation was performed to find optimal hyperparameters. For each conventional classification algorithm, every combination of grid values shown in Table A3 was evaluated by using the complete data set of 14,530 samples and the entire MWIR feature subset. The subset was chosen because the MWIR data already scored high F1-Scores (0.973), compared to the combination of MWIR and NIR (0.978). Therefore, a feature space with fewer dimensions was chosen to prevent over-fitting. The deep learning models were trained using the augmented welding data set, consisting of 72,650 samples of raw image data and image sequences respectively (see Section 3.3). In Figure 9, the performance and the optimal hyperparameter of all classification methods evaluated during grid search are shown.

Overall, the proposed ensemble CNN-GRU architecture achieves the highest classification scores (0.995) and the lowest score variance. However, conventional classification methods such as kNN and non-linear SVM, which are based on geometric and statistical features extracted from the MWIR images, are only slightly lower in terms of their median performance scores. The results show that the average performance level of all methods investigated is high (>90%), which leads to rather small differences between the individual methods. Interestingly, the established and pretrained deep learning model did not reach the level of the proposed ensemble CNN-GRU. This could be due to the large difference between the original training dataset (ImageNet) and the current weld dataset, as well as suboptimal learned features with respect to the new recognition task.

The algorithms can be evaluated not only according to their prediction performances, but also in terms of individual training and inference times, which are particularly important in practice for real-time measurement and quality prediction. If training and inference times are important, the conventional classifier underperforms in contrast to its deep learning competitors. The main reason is the high computational cost for the image processing pipeline, which requires image-wise calculation of geometric and statistical features. Feature calculation runs on the CPU and takes an average of 13.68 ms per image, which limits the maximum FPS to 73 images/sec if the classification time is neglected. In contrast, the trained ensemble CNN-GRU architecture reaches 276 images/sec when inference is performed on GPU/CPU without further optimization. All described models and algorithms were trained on a computer with Intel^®^ Core™ i7-9700 CPU and Nvidia^®^ GeForce^®^ GTX 1080 Ti GPU.

It should be noted that the generally high level of performance of all algorithms may be due to the conservative annotation process of the weld data. Only image data for which the quality of the weld seam could be reliably identified by the human experts were marked accordingly. Therefore, in a next step, we will evaluate the performance of these models on complete and unseen welding trials.

### 4.3. Experimental Evaluation

Four different welding trials were employed to assess the performance of the different classification models. The probability curves of each defect class predicted by the individual classification heads of the ensemble CNN-GRU (i.e., kNN, SVM, fully connected layers), are shown in Figure 10, Figure 11, Figure 12 and Figure 13 for different welding trials, together with their cross-sectional and top view. A description of the welding trails and the applied process parameter can be found in Table 4.

In weld 42 (Figure 10), the welding speed was temporally reduced to 75% of the original welding speed of 3 m/min, which leads to an increased width weld seam that was sufficiently detected by all classifiers and accordingly to the resulting ensemble classification. Additionally, open pores occurred during the weld at the marked positions (red circles) in the top view of weld 42. While the classifiers were not trained to detect this kind of defect, the ensemble model shows high sensitivity to these events, as it presents decreased probabilities for a good weld for these specific seam positions (red circle). In Figure 11, the prediction results for weld 46 are shown. Based on the experimental setup in Figure 2, weld defects were provoked by modifications in the form of several slots at the top side of the middle sheet. In the top view, seam collapses and sagging can be observed (blue circles).

Lack of fusion (red circles) occurred at three positions correctly identified by the classifiers. A short section after frame 1550 was predicted as sagging followed by lack of fusion by the ensemble. However, this cannot be confirmed in the cross-sectional view of the weld.

In Figure 12, the prediction results for weld 48 are shown. In this weld, slots were made on the underside of the middle sheet to induce welding defects. While the top view of the weld shows a small section where sagging occurred, the cross-sectional view shows three sections of lack of fusion defects.

The latter defect type was correctly predicted in terms of their general location, but the exact position was not perfectly recognized. The sagging defect in the first part of the weld seam is detected by the original classification head (NN) and the kNN model as part of the CNN-GRU, which leads to a sagging classification by the ensemble at this location. In Figure 13, the bottom view shows lack of penetration for the entire weld. The welding seam was performed with a laser beam that was positioned -2 mm out of focus. At the end of the weld, the bottom view shows an increased penetration depth. However, full penetration was never achieved during this weld. All models predict the absence of a sufficient weld depth in the first part of the weld with a high probability. In the last third of the weld, according to all classifiers, the probability for lack of penetration decreases. The performance of all classification models used in this work can be obtained from Table 5. Based on four welding trials, the proposed ensemble deep learning architecture achieves an average F1 score of 95.2%, outperforming all other models and showing high robustness and error detection performance for the four welding trials.

It should be noted that in this evaluation all major defects were properly identified by the ensemble classifier. The inaccuracy is due to imprecise localization and dimensions of the defect predictions as well as false positive detections (false alarms) at some points of the weld.

Comparing the individual classical learning algorithms with the proposed architecture, the kNN and SVM classifiers achieve the highest and second highest accuracy among the classical algorithms. This is one argument why these algorithms were considered as part of the ensemble architecture. Overall, the results indicate that the proposed ensemble deep learning architecture achieves the highest classification performance. It is assumed that the performance of classical ML methods and deep learning can be further improved by increasing the amount of training data. However, deep learning methods can also learn to extract more refined features from larger data sets, while traditional methods may reach saturation more quickly in terms of classification performance because their ability to improve feature extraction is not inherently given.

It must be mentioned that the present work was realized with data obtained in a well-controlled laboratory setup. Although the welding head and monitoring equipment studied in this work can also be used for industrial production, the artificially induced faults may not fully reflect situations in industrial applications. Another factor to be considered critically is the highly imbalanced data set used for training and testing. As documented in literature, highly imbalanced data sets cause heavily biased classification results [65].

This results from the fact that classes with more labeled instances are given more importance than those with far fewer labeled instances, since the classifier’s default learning objective tends to be robust to these minority classes. Therefore, classifiers trained under such a condition tend to categorize the minority classes randomly. In this work, class imbalance was addressed by applying class weights during training for weighting the loss function and considering minority classes more important. However, it is believed that the classification performance of these minority classes can be further improved by addressing the imbalances in the dataset seen in Figure 4 through resampling techniques such as synthetic minority over-sampling [66].

It should also be noted that established quality prediction models only work if the underlying assumptions regarding input/output relationships are not violated. However, changing conditions in the manufacturing environment, e.g., new materials, different suppliers and employees, changing machine conditions, etc., in practice lead to concept drift of the model, which needs to be recognized and mitigated. Concept drift can be counteracted by constantly retraining the model with new data. In addition, active detection of concept drift can be achieved by performing tests to detect changes, e.g., by tracking several statistical properties of the incoming data stream within an adaptive window [67,68].

### 4.4. Real-Time Optimization and Inference Times on Embedded Systems

For deployment in manufacturing environments, the target hardware often represents an embedded system with low energy consumption, small size and industry-compatible thermal design. However, these requirements are typically in conflict with the greater computational resources needed to operate deep learning models.

Therefore, embedded computing boards with integrated GPUs have been developed recently to meet the increasing demands for parallel computing resources on low-power devices. In the following, we use the Nvidia Jetson AGX Xavier SoC device as a target platform to perform real-time inference using the ensemble CNN-GRU model. The platform consists of eight custom Carmel ARMv8.2-A 64-bit CPU Cores and an integrated GPU based on the NVIDIA Volta architecture with 512 CUDA cores and 64 Tensor Cores, which represent programmable fused matrix-multiply-and-accumulate units that execute concurrently alongside CUDA cores for accelerating deep learning inference. 

Table 6 shows the technical specification of the embedded system and a desktop alternative that is also used to train the deep learning models used in this work. In order to improve inference times, the TensorRT (NVIDIA) framework has been utilized.

The framework allows to build a high-performance inference graph for a specific target platform. After implementing the CNN-GRU model in TensorRT, the framework performs layer-specific and platform-specific optimizations and generates the inference engine. These optimizations can include reduction of precision via post-training quantization (e.g., FP32 to FP16), layer and tensor fusion, kernel tuning for target platform, tensor memory optimization and multi-stream execution [69].

The optimized inference times for both hardware architectures can be obtained from Figure 14.

Compared to the TensorFlow implementation in Figure 9, the inference time on desktop GPU (GTX 1080 Ti) with FP32 precision has decreased from 3.6 ms to 0.73 ms, which corresponds to a frame rate of 1369 fps. The mean inference time on the embedded system is determined as 1.10 ms (925 fps) for batch size 1 and in maximal performance mode (MAX-N), which is almost twice the sensor acquisition rate of 500 fps. Only when the embedded system is limited to a power budget of 10 W, the possible frame rate drops to 299 fps. Overall, the results indicate that real-time inference is possible with the proposed ensemble CNN-GRU architecture on the presented embedded GPU system at 30 W. Furthermore, if throughput is more important than latency, the batch size can be increased to 8 (i.e., eight image sequences are processed simultaneously), resulting in 1886 fps instead of 925 fps.

## 5. Conclusions

Based on two different imaging sensors, conventional and deep learning techniques were employed to predict critical weld defects such as lack of fusion (false friends), sagging, irregular seam width and lack of penetration. Methods from the field of computer vision and descriptive statistics were used to extract informative features from the image data recorded during weld processes. An extensive study on the importance of the different features and feature subsets was carried out. It is shown that the most relevant features can be derived from MWIR camera images, especially from the weld pool region, when a small number (<36) of features are used. However, the highest detection rates were achieved by combining geometrical and statistical features extracted from both image data sources. Moreover, an ensemble deep learning architecture based on CNNs, GRUs and high-performance classification algorithms was employed to detect weld defects exploiting their ability to extract spatio-temporal features from raw video data. In a further step, hyperparameters for deep learning methods as well as for classical machine learning algorithms were optimized during an extensive grid search. Compared to all methods investigated in this study, the proposed architecture achieves the best classification results and is able to provide indications of undefined errors such as open pores. Based on the evaluation on four previously unseen welding trials, our proposed architecture achieves the highest mean F1-Score of 95.2% of all investigated classification models and represents a competitive alternative that does not require extensive feature engineering. Finally, inference optimization of the proposed model with respect to an embedded GPU system as target hardware enables the trained model to operate with a processing time of 1.1 ms per input image sequence (i.e., 925 image sequences per second).

In the future, more emphasis will be placed on unsupervised and semi-supervised methods for detecting anomalies and defects using a small number of training samples. Furthermore, it is envisaged to address the imbalances in the datasets, e.g., through cost-sensitive learning or random resampling techniques. In combination with advanced data augmentation methods, this could further increase the performance of the machine learning methods presented in this paper.

## Figures and Tables

**Figure 1 sensors-21-04205-f001:**
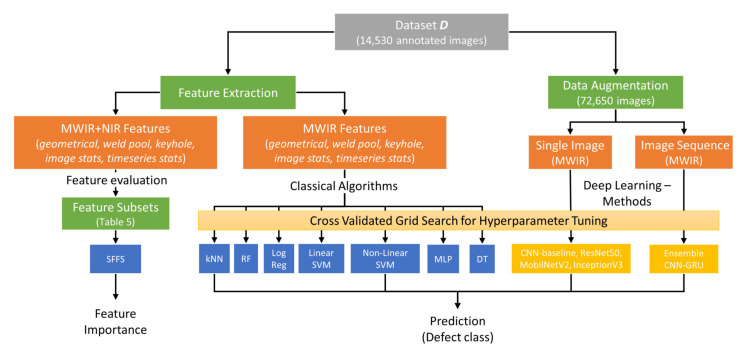
Schematic overview of the evaluation process established in this work.

**Figure 2 sensors-21-04205-f002:**
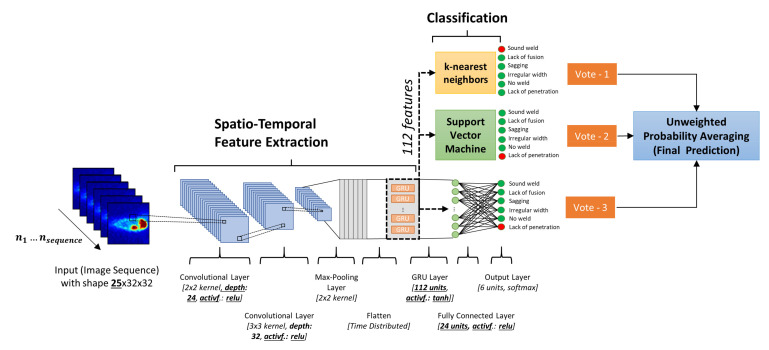
Proposed spatio-temporal ensemble deep neural network architecture based on convolutional layers, gated recurrent units (GRU) and different classification heads for weld defect detection.

**Figure 3 sensors-21-04205-f003:**
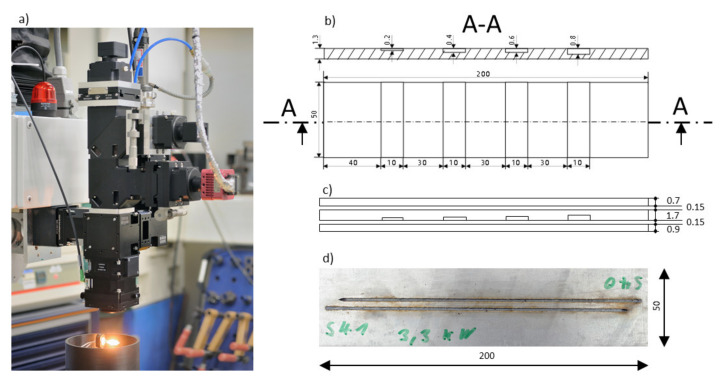
(**a**) Photograph of the welding optics with coaxially integrated cameras; (**b**) Drawing of welding sheets with different slot sizes (middle sheet); (**c**) Side view of the sheet configuration used during the welding experiments; (**d**) Photograph (top view) of two welding trails (P = 3.3 KW, v = 50 mm/s, ds = 0.6 mm, Argon shield gas flow = 60 L/min).

**Figure 4 sensors-21-04205-f004:**
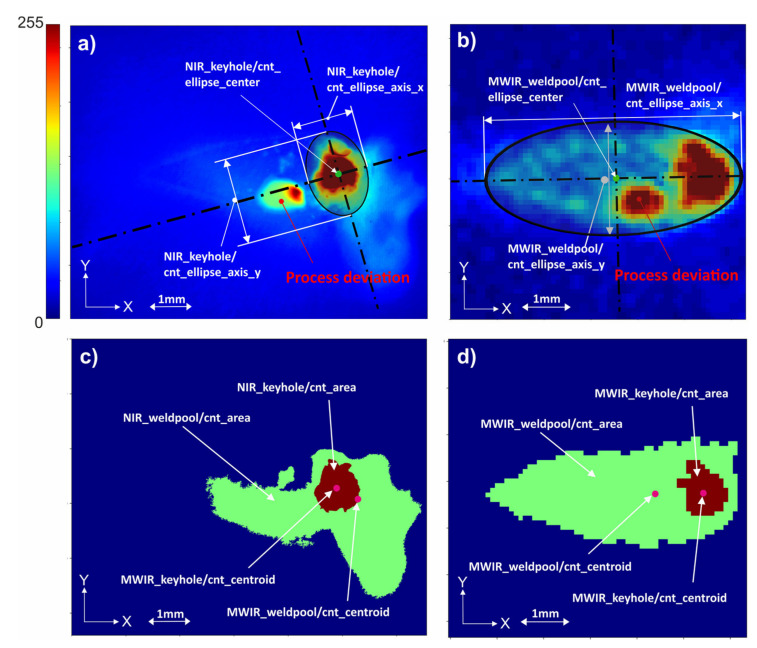
(**a**,**b**) Original image and geometrical features extracted from keyhole and weld pool regions. (**c**,**d**) Detected keyhole and weld pool contours (filled) based on two-step binarization of the original images.

**Figure 5 sensors-21-04205-f005:**
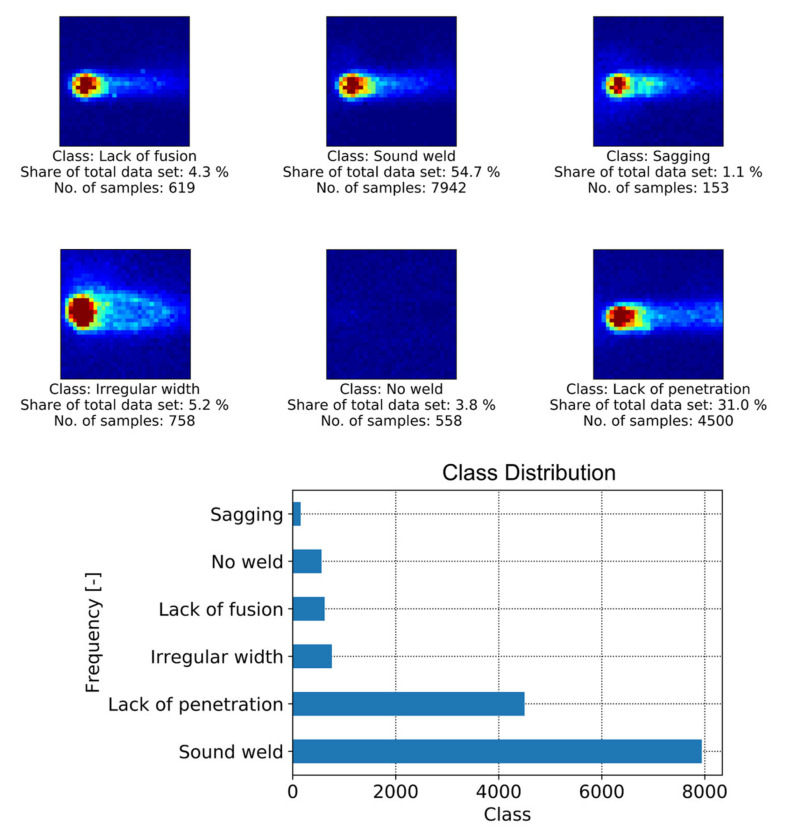
Example of MWIR image data and sample distribution of different quality states based on 13 weld trials (14,530 samples) that form the welding data set.

**Figure 6 sensors-21-04205-f006:**
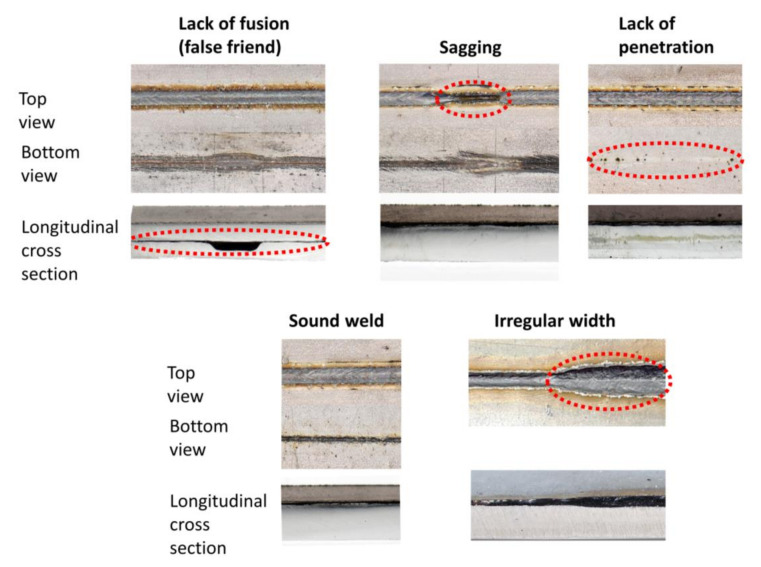
Photographs from different perspectives of the welding defects investigated in this study.

**Figure 7 sensors-21-04205-f007:**
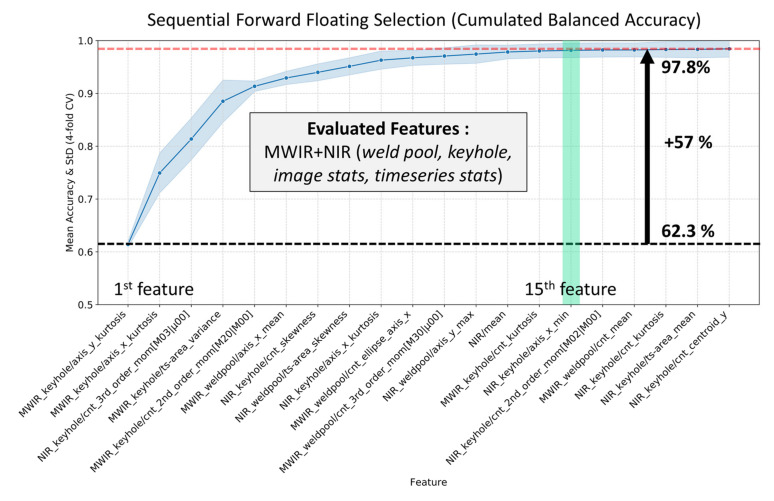
The 20 most important features based on 172 geometrical and statistical characteristics of the weld pool, keyhole and overall image statistics extracted from MWIR images (starting with the left).

**Figure 8 sensors-21-04205-f008:**
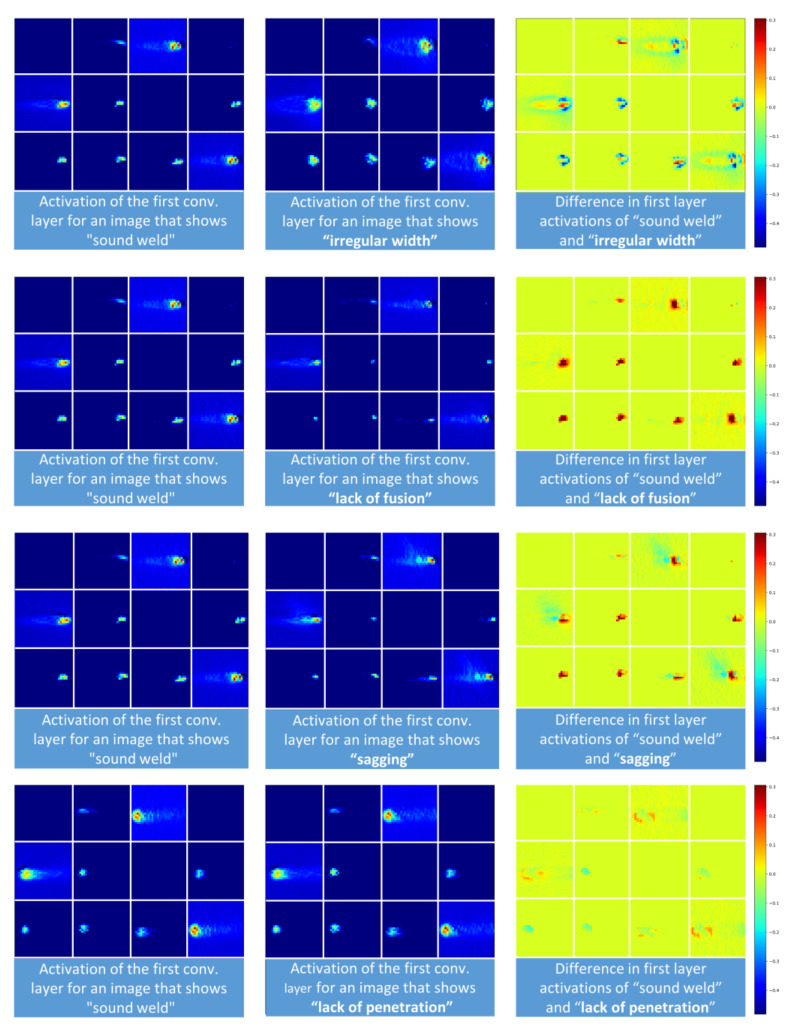
Layer activations based on 12 different filter kernels in the first layer of a CNN that was trained to recognize welding defects based on images from the MWIR camera. Each square shows a single activation map.

**Figure 9 sensors-21-04205-f009:**
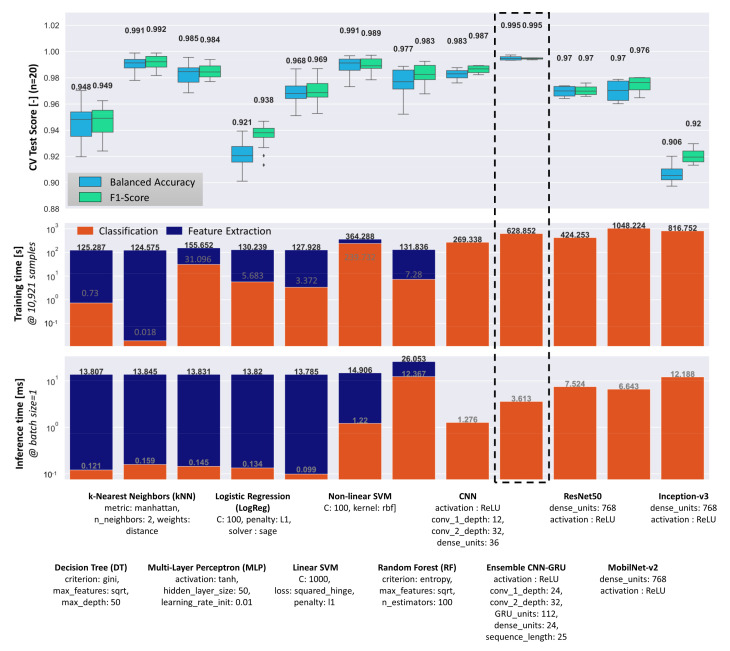
Performance comparison of different conventional machine learning and deep learning classification methods. Optimal hyperparameter for each classifier were found via grid search (Table A3). The median scores are displayed in the top diagram.

**Figure 10 sensors-21-04205-f010:**
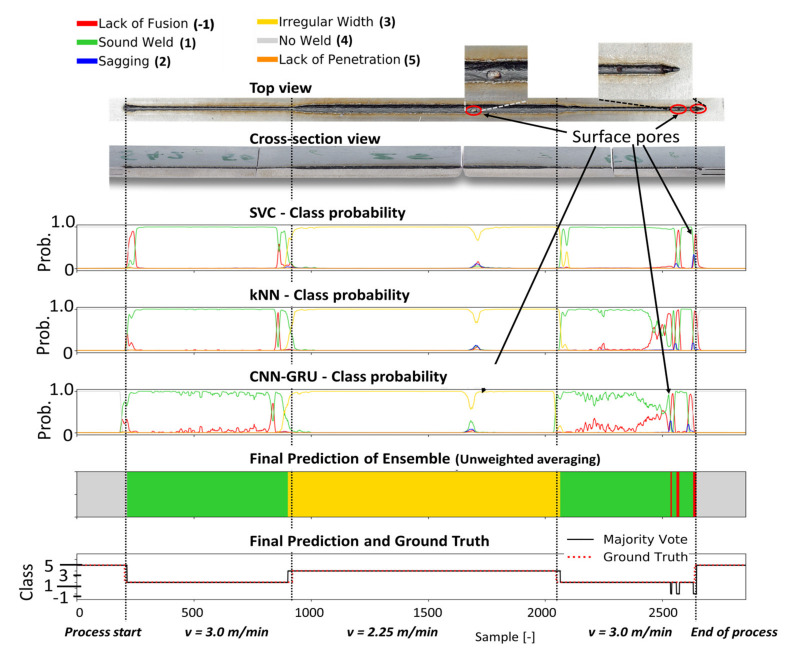
The metallographic characterization, the resulting ground truth data and the classification results for weld 42 based on the proposed ensemble CNN-GRU architecture.

**Figure 11 sensors-21-04205-f011:**
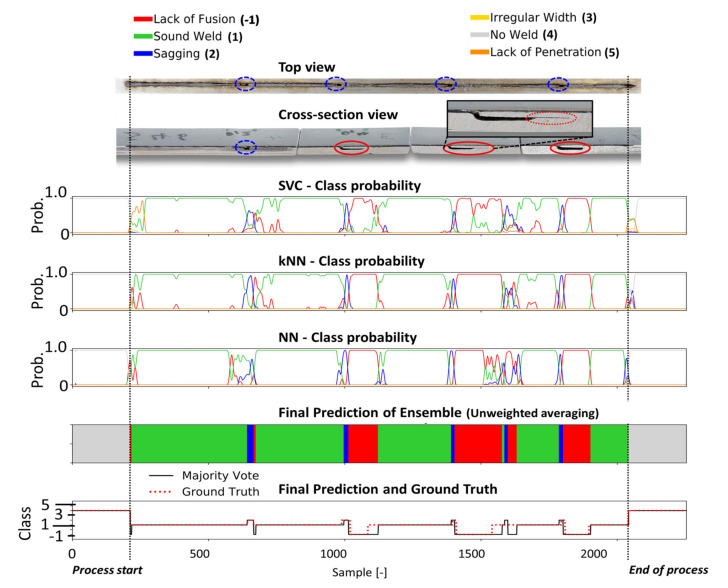
The metallographic characterization, the resulting ground truth data and the classification results for weld 46 based on the proposed ensemble CNN-GRU architecture.

**Figure 12 sensors-21-04205-f012:**
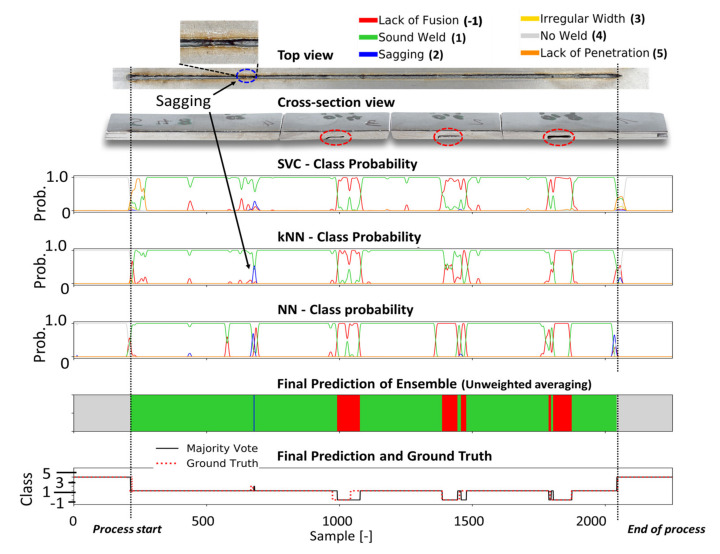
The metallographic characterization, the resulting ground truth data and the classification results for weld 48 based on the proposed ensemble CNN-GRU architecture.

**Figure 13 sensors-21-04205-f013:**
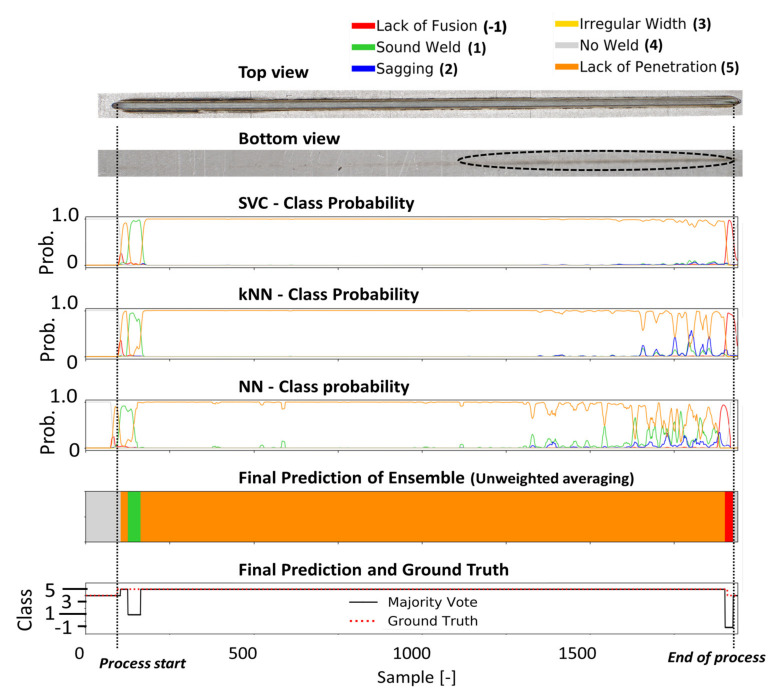
The metallographic characterization, the resulting ground truth data and the classification results for weld 216 based on the proposed ensemble CNN-GRU architecture.

**Figure 14 sensors-21-04205-f014:**
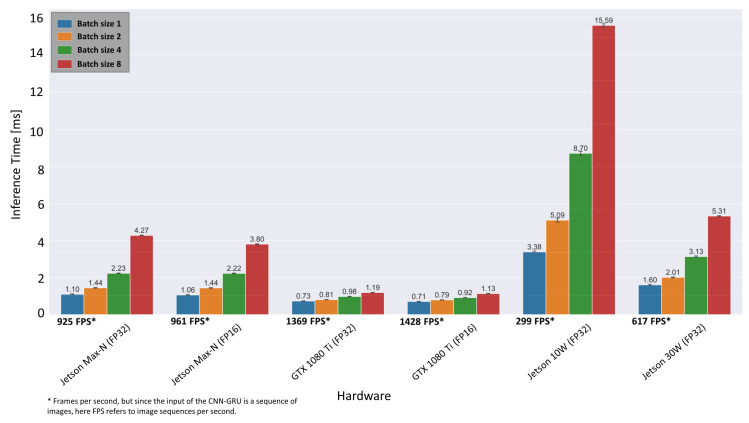
Inference results of the CNN-GRU architecture after optimization via TensorRT for different hardware setups.

**Table 1 sensors-21-04205-t001:** Description of the sensors and optical components used for the welding experiments.

Type of Camera	Sensor Material/ Sensitivity Range	Resolution	Acquisition Rate	Field of View	Bandpass Filter (CWL/FWHM)	Interface
Photonfocus D1312IE-160-CL (NIR)	Si/0.4–0.9 µm	1312 × 1080	100 Hz	11.6 × 5 mm^2^	840 nm/40 nm	CameraLink
NIT Tachyon μCore 1024 (MWIR)	PbSe/1–5 µm	32 × 32	500 Hz	9 × 9 mm^2^	1690 nm/82 nm	USB2.0

**Table 2 sensors-21-04205-t002:** Description of feature sub-groups used for classical machine learning methods and feature importance evaluation.

Feature Sub-Group (Short Name)	Expression	Description
Geometrical features (*geometrical*)	$G_{i}^{T}$	Only geometrical features according to Table A1 based on the weld pool and keyhole region
Overall image statistics (*image stats*)	$IS_{i}^{T}$	Overall image statistics according to Table A2
Times series statistics (*timeseries stats*)	$TS_{i}^{T}$	Time series statistics according to Table A2 based on weld pool area
Weld pool features (*weld pool*)	$WP_{i}^{T}$	Geometrical and statistical features according to Table A1 and Table A2 derived from the weld pool region
Keyhole features (*keyhole*)	$KH_{i}^{T}$	Geometrical and statistical features according to Table A1 and Table A2 derived from the keyhole region

**Table 3 sensors-21-04205-t003:** Comparison of several feature subsets with respect to their ability to predict different weld defects (without “no weld” class).

Feature Subset		Cross-Validated F1-Score					
Name	No. of Feat.	Lack of Fusion	Sound Weld	Sagging	Irregular Width	Lack of Penetration	Avg
MWIR+NIR (*weld pool, keyhole, image stats, timeseries stats*)	172	0.983	0.998	0.913	1.0	0.999	0.978
MWIR+NIR (*geometrical*)	64	0.89	0.989	0.976	1.0	0.998	0.970
MWIR+NIR (image stats)	12	0.743	0.908	0.091	0.999	0.951	0.738
MWIR+NIR (*timeseries stats*)	12	0.701	0.867	0.000	1.0	0.914	0.694
MWIR+NIR (*weld pool*)	74	0.953	0.995	0.901	1.0	0.999	0.969
MWIR+NIR (*keyhole*)	74	0.948	0.995	0.829	1.0	0.999	0.954
MWIR (*weld pool, keyhole, image stats, timeseries stats*)	86	0.945	0.993	0.93	1.0	0.998	0.973
MWIR (*geometrical*)	32	0.834	0.96	0.864	1.0	0.98	0.928
MWIR (*image stats*)	6	0.688	0.74	0.000	1.0	0.862	0.658
MWIR (*timeseries stats*)	6	0.569	0.669	0.000	1.0	0.801	0.607
MWIR (*weld pool*)	37	0.896	0.987	0.951	1.0	0.997	0.966
MWIR (*keyhole*)	37	0.851	0.983	0.833	1.0	0.997	0.933
NIR (*weld pool, keyhole, image stats, timeseries stats*)	86	0.904	0.986	0.956	1.0	0.996	0.968
NIR (*geometrical*)	32	0.56	0.907	0.780	0.923	0.961	0.826
NIR (*image stats*)	6	0.403	0.862	0.000	0.922	0.937	0.625
NIR (*timeseries stats*)	6	0.544	0.808	0.000	0.989	0.855	0.639
NIR (*weld pool*)	37	0.787	0.941	0.902	0.993	0.971	0.918
NIR (*keyhole*)	37	0.791	0.955	0.863	0.995	0.978	0.916

**Table 4 sensors-21-04205-t004:** Parameters and sheet configuration of four welding experiments for evaluation.

Welding Parameters	Weld 42	Weld 46	Weld 48	Weld 216
Laser power (kW)	3.3	3.3	3.3	2.7
Beam focus offset (mm)	0	0	0	−2
Welding speed (mm/s)	50; 37.5	50	50	50
Shielding gas (L/min)	60	60	60	60
Sheet configuration	Three sheets; No slots	Three sheets; Slots point upwards	Three sheets; Slots point downwards	Two sheets; No middle sheet

**Table 5 sensors-21-04205-t005:** Classification performance for different welding trials (not within the training data set).

Method	Weld 42 (2856 Samples)	Weld 46 (2255 Samples)	Weld 48 (2254 Samples)	Weld 216 (3140 Samples)	Avg. Accuracy	Avg. F1-Score
Decision Tree	0.893	0.861	0.914	0.729	0.849	0.893
kNN	0.977	0.885	0.921	0.94	0.931	0.939
MLP	0.962	0.882	0.916	0.831	0.898	0.924
LogReg	0.944	0.873	0.911	0.782	0.878	0.892
Linear SVM-	0.93	0.815	0.906	0.75	0.85	0.867
Non-Linear SVM *	0.958	0.892	0.917	0.888	0.914	0.926
RF	0.97	0.921	0.927	0.796	0.904	0.923
CNN-baseline	0.822	0.895	0.916	0.933	0.892	0.897
ResNet50	0.91	0.823	0.894	0.9174	0.887	0.902
MobileNetV2	0.9488	0.869	0.9041	0.965	0.922	0.922
InceptionV3	0.967	0.898	0.919	0.821	0.908	0.905
Ensemble CNN-GRU *	0.973	0.923	0.944	0.963	0.951	0.952

* Proposed method.

**Table 6 sensors-21-04205-t006:** Overview of the processing hardware used in this study.

Hardware	NVIDIA GeForce GTX 1080 Ti	NVIDIA JETSON AGX XAVIER
Type	Desktop GPU	Embedded GPU SoC
Power Consumption (TDP)	250 Watt	10W/15W/30W/Max-N * profiles
Cuda Cores	3584	512
Memory	11GB (dedicated)	16GB (Shared)
Clock Speed (Base/Boost))	1.48 GHz/1.58 GHz	0.85 GHz/1.37GHz
Memory Bandwidth	484.4 GB/s	137 GB/s
Size	267 mm × 112 mm (card only)	87 mm × 100 mm

* Max-N mode (~50 W): no cap for power budget, maximum CPU, GPU and memory frequencies; Software: cuDNN 8.0/TensorRT 7.1.3.

## Data Availability

Data presented in this study can be obtained from the corresponding author upon request.

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
