# Peer review of "A Spatio-Temporal Ensemble Deep Learning Architecture for Real-Time Defect Detection during Laser Welding on Low Power Embedded Computing Boards"

_sensors, 2021, doi:10.3390/s21124205_

Round 1
Reviewer 1 Report
Dear authors, congratulations on the work done.
Technical content is OK. Nothing to add or modify.
In my opinion, you only need to make a detailed review on some format aspects, as messages like "Error! Reference source not found." and some confusion in numbering Tables and Figures through the text.
Table 8 is not called in the text. I suggest you refer to it at the beginning of section 4.3 to clarify to readers what to expect when seeing Figures 12 and 13.
Also, make some improvements in figures' text sizes. Sometimes they are quite small (for example in figures 7, 12, and 13). More than this, Figures 13 and 13 deserve amplification of weld beads pictures. They are key to understand the results.
One last suggestion is to put some paragraph on conclusions about what to expect when dealing with different situations as welding in regular production (in a regular batch form) or with different materials and what to concern about when adjusting the sensor to that situation. Feel free to accept this or not. It is just a suggestion.
Best regards!
Reviewer 2 Report
Fault detection is investigated for laser welding using deep learning and convolutional neural networks. The topic is interesting, and the presentation and organisations are done well. The paper can be published subjected to minor revisions. The followings would benefit the improvement of the quality of the paper.
1) Immediately after (9), why can Tmelt=1737K lead to a maximum thermal radiance from the viewpoint of theory analysis?
2) In (11), all the components have the same weights. The reasonableness should be explained.
3) The contribution of the paper seems to lie in the application of the algorithms. The innovation of the paper should be better highlighted.
4) Some recent relevant special issues on fault diagnosis and monitoring would help enhance the research motivation of the paper, e.g., ``Real-time fault diagnosis and fault tolerant control''; "Advances in Condition Monitoring, Optimization and Control for Complex Industrial Processes".
5) The relevance of the paper to the journal would need to be strengthened, as none of the references come from this journal. No relevant papers were published in this journal before?
To summarize, this is good-quality paper, which can be published subjected to minor revisions.
Reviewer 3 Report
- As can be seen from Fig. 11, when the accuracy of the proposed CNN-GRU is improved by 0.001 compared with that of CNN, the training time is increased by about 623s and the reasoning time is increased by 1s, which is undoubtedly not conducive to online detection. In this case, is the proposed method more suitable for the process monitoring mentioned in the abstract?
- How does the proposed deep learning framework compare with advanced image classification frameworks such as AlexNet and ResNet?
- What are the advantages and disadvantages of the proposed ensemble classifier? Do the disadvantages outweigh the advantages when used for process monitoring?
- After confirming the higher accuracy of CNN-GRU network, the author integrated it with traditional machine learning methods. Is this in conflict with the contrast between deep learning and traditional methods emphasized in the title?
- The paper describes the existing methods in a longer length, and it is better to highlight its own innovation and research ideas.
- The quality of Figure 1 can be improved. It is recommended to check the English expression of the whole text and try to avoid the expression of Chinglish.
Reviewer 4 Report
It is an original work that presents a novel approach for in-situ fault detection during laser welding by means of the use of classical machine learning methods, deep learning algorithms, and feature extraction and selection techniques. This approach is very interesting from a scientific point of view, but its real practicability is not enough clear. There is at least one note, which authors must take into consideration:
Welding is the manufacturing process and the use of complicated methods of the detection of defects must it should be justified from an economic point of view. For this reason, authors must clearly describe those fields of application of the welding technology where the proposed approach is really needed.
Round 2
Reviewer 3 Report
Based on two different imaging sensors, the paper uses conventional and deep learning techniques to predict critical weld defects such as lack of fusion (false friends), sagging, irregular seam width and lack of penetration. The proposed method has certain application value. However, the quality of this paper will be improved if the author can describe the difficulties in more detail which is encountered in studying the method and how to solve them.
